# Serum biomarkers and anti-flavivirus antibodies at presentation as indicators of severe dengue

Cynthia Bernal[1], Sara Ping[2], Alejandra Rojas[1], Oliver Caballero[1], Victoria Stittleburg[2], Yvalena de Guillén[1], Patricia Langjahr[3], Benjamin A. Pinsky[4,5], Marta Von-Horoch[6], Patricia Luraschi[6], Sandra Cabral[6], María Cecilia Sánchez[7], Aurelia Torres[7], Fátima Cardozo[1,7‡]*, Jesse J. Waggoner[2,8‡]*

1 Universidad Nacional de Asunción, Instituto de Investigaciones en Ciencias de la Salud, San Lorenzo, Paraguay, 2 Emory University School of Medicine, Division of Infectious Diseases, Atlanta, Georgia, United States of America, 3 Universidad Nacional de Asunción, Facultad de Ciencias Químicas, San Lorenzo, Paraguay, 4 Department of Pathology, Stanford University School of Medicine, California, United States of America, 5 Department of Medicine, Division of Infectious Diseases and Geographic Medicine, Stanford University School of Medicine, California, United States of America, 6 Hospital Central—Instituto de Previsión Social, Departamento de Epidemiología, Asunción, Paraguay, 7 Hospital Central—Instituto de Previsión Social, Departamento de Laboratorio de Análisis Clínicos, Asunción, Paraguay, 8 Rollins School of Public Health, Department of Global Health, Atlanta, Georgia, United States of America

☯ These authors contributed equally to this work.
‡ FC and JJW also contributed equally to this work.
* fati.cardozo@hotmail.com (FC); jjwaggo@emory.edu (JJW)

**Data Availability Statement:** All relevant data are within the manuscript and its Supporting Information files.

## Abstract

### Background

Dengue is the most common vector-borne viral disease worldwide. Most cases are mild, but some evolve into severe dengue (SD), with high lethality. Therefore, it is important to identify biomarkers of severe disease to improve outcomes and judiciously utilize resources.

### Methods/Principal findings

One hundred forty-five confirmed dengue cases (median age, 42; range <1–91 years), enrolled from February 2018 to March 2020, were selected from an ongoing study of suspected arboviral infections in metropolitan Asunción, Paraguay. Cases included dengue virus types 1, 2, and 4, and severity was categorized according to the 2009 World Health Organization guidelines. Testing for anti-dengue virus IgM and IgG and serum biomarkers (lipopolysaccharide binding protein and chymase) was performed on acute-phase sera in plate-based ELISAs; in addition, a multiplex ELISA platform was used to measure anti-dengue virus and anti-Zika virus IgM and IgG. Complete blood counts and chemistries were performed at the discretion of the care team. Age, gender, and pre-existing comorbidities were associated with SD vs. dengue with/without warning signs in logistic regression with odds ratios (ORs) of 1.07 (per year; 95% confidence interval, 1.03, 1.11), 0.20 (female; 0.05, 0.77), and 2.09 (presence; 1.26, 3.48) respectively. In binary logistic regression, for every unit increase in anti-DENV IgG in the multiplex platform, odds of SD increased by 2.54

**Funding:** Research was supported by the Doris Duke Charitable Foundation (Clinical Scientist Development Award 2019089 to JJW); National Institute of Allergy and Infectious Diseases (R21 AI146443 to AR, BAP, and JJW); and the Consejo Nacional de Ciencia y Tecnología (CONACYT) with support from the Fondo para la Excelencia de la Educación y la Investigación, both in Paraguay (PINV18-1295 to AR and POSG17-59 to CB). The funders had no role in study design, data collection and analysis, decision to publish, or preparation of the manuscript.

(1.19–5.42). Platelet count, lymphocyte percent, and elevated chymase were associated with SD in a combined logistic regression model with ORs of 0.99 (1,000/µL; 0.98,0.999), 0.92 (%; 0.86,0.98), and 1.17 (mg/mL; 1.03,1.33) respectively.

## Conclusions

Multiple, readily available factors were associated with SD in this population. These findings will aid in the early detection of potentially severe dengue cases and inform the development of new prognostics for use in acute-phase and serial samples from dengue cases.

## Author summary

Dengue fever is an acute disease caused by dengue virus and transmitted to humans through the bite of infected *Aedes* mosquitoes. Dengue is the most common vector-borne viral disease worldwide affecting an estimated 50–100 million people and with 10,000 dengue-related deaths each year. Currently, there is no specific treatment, and safe and effective vaccines have not been fully implemented. Most dengue cases present with nonspecific mild symptoms, but some will evolve into severe dengue, which can be fatal. Early detection and subsequent timely treatment have been shown to decrease mortality among severe cases. Therefore, it is very important to identify biomarkers for the early identification of cases at risk for progression to severe disease. In this study we analyze demographic factors, clinical laboratory data, lipopolysaccharide binding protein and chymase to evaluate associations with disease severity. This study was carried out in Paraguay, which is a hyperendemic country for dengue where the disease has been understudied. A number of factors were found to be associated with severe disease in this population, including patient age, male gender, presence of comorbid illnesses, low platelet count, low lymphocyte percentage, and elevated chymase level.

## Introduction

Dengue is a common acute febrile illness in tropical and subtropical regions of the world and accounts for upwards of 10% of such illnesses in areas of endemicity [1–4]. Over the past 30 years, dengue incidence and associated deaths have increased both in those residing in and travelers returning from areas of endemicity [5, 6]. An estimated 50–100 million dengue cases and 10,000 dengue-related deaths annually occur worldwide from infection with one of the four types of dengue virus (DENV, genus *Flaviviridae*) [1, 3–5, 7]. Dengue severity ranges dramatically from a mild subclinical illness to dengue fever and clinically severe dengue with plasma leakage, hemorrhage, and/or end-organ dysfunction [1, 3, 4, 8, 9]. Timely diagnosis and the initiation of appropriate supportive care improves clinical outcomes and can lower mortality in clinically severe dengue from 20% to <1% [1, 3, 4, 10, 11]. Although clinically severe cases represent a minority of dengue cases overall, fatal and hospitalized non-fatal cases account for over half of the $8.9 billion USD annual economic burden of dengue [9, 12]. Therefore, early identification of cases at increased risk for developing clinically severe dengue could both improve clinical outcomes and alleviate the economic burden caused by dengue on resource constrained medical systems [13].

Clinically severe dengue results from a complex interplay of virus [14–17], host [18–25], and epidemiologic factors [1, 9, 18]. The manifestations of severe dengue also differ based on

patient age, with children more commonly developing plasma leakage compared to hemorrhage in adults [9, 26, 27]. Studies have identified associations between the detection and/or concentration of various molecules or gene transcripts and severe dengue [28–33]. One group of biomarkers that has been studied are proteins released during mast cell degranulation: vascular endothelial growth factor (VEGF) and the proteases tryptase and chymase [34–44]. In studies of patients from South and Southeast Asia, chymase was associated with and predictive of the development of clinically severe dengue [34, 35, 37, 38, 41]. Chymase release from mast cells occurs in the presence of DENV and may be increased by pre-existing anti-DENV IgG antibodies [36, 37]. According to a single study in mice, chymase release may be blocked by antibodies against viral non-structural protein 1 (NS1) [45]. Lipopolysaccharide (LPS) and lipopolysaccharide binding protein (LBP) are another set of molecules that have higher levels in dengue cases compared to healthy controls and in clinically severe cases compared to dengue fever, which could indicate their usefulness as a predictor of severity [46–48]. Elevated levels of circulating LPS and LBP result from derangements in gut permeability, potentially leading to bacterial translocation, bacteremia, and worsened outcomes. Finally, numerous clinical laboratory findings have been associated with clinically severe dengue, such as thrombocytopenia, lymphopenia, and evidence of liver or kidney injury [16, 20, 27, 49–51]. These may either define cases as clinically severe with end-organ dysfunction or predict the development of severe dengue through detection of changes over the course of illness [3].

The objective of the current study was to evaluate biomarkers of dengue severity among participants enrolled in an ongoing study of acute arboviral illness in the metropolitan area of Asunción, Paraguay. Paraguay is hyperendemic for dengue, with sustained viral circulation since 1999 and large disease outbreaks occurring every 2–5 years. In 2018, predominant circulation of DENV-1 was recorded [52], and in 2019–2020, this shifted to DENV-4, resulting in the largest outbreak in the country's history [53]. Previous studies from Paraguay have found an increased risk of clinically severe dengue with DENV-2 and secondary infections [54–56]. However, dengue, and in particular biomarkers of severe disease, remains understudied in the country [57, 58]. Previously, our group evaluated anti-DENV and anti-ZIKV NS1 IgG levels among dengue cases in 2018 using a multiplex serological assay, the pGOLD assay [59]. Anti-DENV IgG levels in the pGOLD assay correlated with focus reduction neutralization test (FRNT50) titers, and an association was observed between hospitalization and detection of both anti-DENV and anti-ZIKV IgG. However, hospitalization is an inexact measure of clinical dengue severity. Therefore, in the current study, we sought to evaluate this earlier finding and levels of chymase and LBP as indicators of dengue severity among participants categorized according to the 2009 World Health Organization guidelines [3].

## Methods

### Ethics statement

The study protocol was reviewed and approved by the Scientific and Ethics Committee of the Instituto de Investigaciones en Ciencias de la Salud, Universidad Nacional de Asunción (IIC-S-UNA, IRB00011984), and the Emory University Institutional Review Board (IRB00000569). Written informed consent was obtained from all the participants or their health care decision maker.

### Clinical samples

Individuals included in the current study were enrolled in an ongoing parent study of suspected arboviral infections in the Asunción metropolitan area between February 2018 and March 2020. Participants of both genders and all ages were enrolled as outpatients at

IICS-UNA in all study years and in the emergency care/inpatient facilities of Hospital Villa Elisa, 2018, and Hospital Central of the Instituto de Previsión Social, 2019–2020. Inclusion criteria for the parent study were an acute illness including two or more of the following symptoms: fever (measured or subjective), red eyes, rash, joint pain involving more than one joint, and/or diffuse muscle pain. Patients with fever and no other localizing signs or symptoms were also included. Day 1 was defined as the first day of symptoms.

One hundred forty-five participants with acute dengue and up to 7 days of symptoms were selected for the current cross-sectional analysis from a total study population of 1566 cases of suspected arboviral illness. Cases were classified according to the 2009 WHO criteria as dengue without warning signs (DWS-), dengue with warning signs (DWS+) and severe dengue (SD) [3]. Cases were classified during the initial visit, and the final classification used for this study was upgraded if the case evolved over time to a more severe category following presentation. For categorization as DWS+, it was necessary to have at least one warning sign. For categorization as SD, an individual had to develop at least one criterion for SD during the clinical course. To maximize study power, all SD cases in the parent study were included in this analysis. A mixture of DWS- and DWS+ cases was then selected to achieve a representative distribution of participants based on age, days of symptoms, comorbidities, and gender from across the study period and to maintain an even distribution of these two categories. The number of included cases was limited by sample volume and availability of demographic and clinical data.

## Laboratory testing

Acute-phase serum or plasma samples were collected during the initial visit for study enrollment and transported to the IICS-UNA laboratory. Samples were tested for DENV NS1 antigen using the Standard Q Dengue Duo rapid immunochromatographic test (SD Biosensor, Suwon, South Korea) according to manufacturer recommendations. Qualitative antibody data acquired using this method was not evaluated in this study, see antibody section below. Primary samples were then aliquoted and stored at −80°C until later use or shipment on dry ice to Emory University for additional testing. For molecular testing, total nucleic acids were extracted from 200μL of sample on an EMAG instrument and eluted into 50μL of buffer. Samples were tested for Zika virus, chikungunya virus and DENV by real-time RT-PCR (rRT-PCR) using a validated and published multiplex assay (the ZCD assay) [60], and DENV serotype and viral load were determined with a published DENV multiplex assay [61, 62]. Both rRT-PCRs were performed as previously described [60–62].

Serologic testing was performed on acute-phase samples using two different methods. First, anti-DENV IgG and IgM were analyzed using commercial ELISA kits [Dengue ELISA IgG (G1018) and Dengue ELISA IgM Capture (M1018), Vircell Microbiologists, Granada, Spain] according to manufacturer recommendations (interpretation: IgM or IgG index >11 positive, 9–11 indeterminate, <9 negative). Second, a 5μL aliquot of serum from 139 participants with sufficient sample was tested in the pGOLD assay (Nirmidas Biotech, Inc, Palo Alto, CA), which is a multiplex serological assay for IgM and IgG against DENV (DENV-2 whole virus antigen) and ZIKV (NS1 antigen). The pGOLD assay was performed as previously described [59, 63]. In each well of the pGOLD slide, antigens are spotted in triplicate, and average signals are used during analysis. For IgG, the negative control signal was subtracted from the sample signal, and the difference was divided by the average signal of four IgG control spots included in each well. For IgM, a similar calculation was performed using the signal from a known anti-DENV IgM positive control sample included on each run. A positive threshold ratio of 0.1 was established for each isotype, which was ≥ 3 standard deviations above the mean of the negative control.

Chymase and LBP levels were determined using commercial ELISA kits (G-Biosciences, St. Louis, MO, USA), following the manufacturer's instructions. Complete blood counts and chemistries were performed at the clinical site at the discretion of the care team, and results were included if the sample was obtained within ±1 day of enrollment.

### Case definitions

Dengue cases were defined as individuals who met inclusion criteria for the parent study and had 1) detectable DENV RNA in the ZCD and/or DENV multiplex rRT-PCR or 2) detection of DENV NS1 by rapid test. For a single participant with DWS-, dengue was defined based on clinical presentation and a strong epidemiologic during a large was of DENV-4 cases.

### Statistical analysis

Basic statistical analyses were performed using Excel software (Microsoft, Redmond, WA). Comparisons between group means and medians were made by the ANOVA, Welch's test, both pooled and non-pooled two sample t-tests, Mann-Whitney U test, and Kruskal Wallis tests. Comparisons of proportions were made using chi-squared tests or Fisher exact tests (if the expected number in each cell was <5). Graphs were prepared with GraphPad Prism version 9 (GraphPad, San Diego, CA). Crude associations, statistical analysis and modeling were performed using SAS version 9.4. To calculate odds ratios for SD, domain models were developed using demographic (age, gender, comorbidities) and laboratory variables (basic clinical laboratory results, DENV viral load, chymase and LBP). Models were evaluated using binomial logistic regression (DWS-/DWS+ vs. SD), and goodness of fit was evaluated using area under the receiver operating curve (AUROC). Significance was set at two-sided p-value ≤0.05 for all analyses. Comorbidities were defined as present or absent for all statistical analysis.

## Results

### Demographic and clinical information

Of 145 participants in this study, 55 were categorized as DWS-, 67 as DWS+, and 23 as SD. Demographic data and DENV diagnostic test results are shown in Table 1 (binary categories) and S1 Table (three categories). Participants were enrolled primarily at Hospital Central of the Instituto de Previsión Social (n = 124), followed by Hospital Villa Elisa (15) and IICS-UNA (6). Results for DWS- and DWS+ were not significantly different for most analyses performed in this study. As such, results are reported for analyses using the binary outcome of DWS-/DWS+ vs. SD, except where indicated. Data and analyses for the three individual categories are provided in the Supplemental Material.

SD cases were significantly older than non-SD cases and were significantly more likely to be male and have at least one comorbidity (Table 1). The presence of specific comorbidities also differed by population (Table 1). In logistic regression of these variables in relation to disease severity, age, gender, and comorbidities remained in the model and were predictors of severity with a strong goodness of fit (AUROC = 0.94; Table 2). In logistic regression, comorbidity was defined as a discrete variable. In addition, SD cases presented for care later in the course of illness than non-severe cases (Table 1), and more SD cases were included 2020, consistent with the large DENV-4 outbreak that occurred in Paraguay that year [64].

### DENV testing

One hundred forty-four of 145 dengue cases (99.3%) tested positive by rRT-PCR, NS1 rapid test, or both; and only one case was included based epidemiologic criteria alone. Over 90% of

**Table 1. Demographic data and DENV diagnostic test results for participants stratified by dengue severity.**

| Characteristic[a] | DWS-/DWS+ N = 122 | SD N = 23 | p-value |
|---|---|---|---|
| Age, years, mean (st. dev.) | 34 (18) | 61 (19) | <0.001 |
| Gender, female | 81 (66.4) | 6 (26.1) | <0.001 |
| Comorbidities, ≥ 1[b] | 34 (28.1) | 16 (84.2) | <0.001 |
| Hypertension | 21 (17.4) | 11 (57.9) | <0.001 |
| Diabetes | 7 (5.8) | 7 (36.8) | <0.001 |
| Chronic kidney disease | 0 (0) | 3 (15.8) | 0.002 |
| Chronic heart disease | 1 (0.8) | 4 (21.1) | 0.001 |
| Cancer | 2 (1.7) | 0 (0) | 1.00 |
| Autoimmune disease | 8 (6.6) | 1 (5.3) | 1.00 |
| Other | 10 (8.3) | 5 (26.3) | 0.033 |
| Days of symptoms, mean (st. dev.) | 3.9 (1.9) | 4.8 (1.7) | 0.033 |
| Year of Collection | | | 0.015 |
| 2018 | 14 (11.5) | 4 (17.4) | |
| 2019 | 42 (34.4) | 1 (4.3) | |
| 2020 | 66 (54.1) | 18 (78.3) | |
| DENV rRT-PCR, positive | 110 (90.2) | 20 (90.9) | 1.00 |
| Serotype | | | 0.45 |
| DENV-1 | 14 (12.7) | 4 (20.0) | |
| DENV-2 | 9 (8.2) | 0 (0) | |
| DENV-4 | 86 (78.2) | 16 (80.0) | |
| Negative | 1 (0.9) | 0 (0) | |
| NS1, positive | 77 (63.1) | 21 (91.3) | 0.010 |

Abbreviations: st. dev., standard deviation

[a] Presented as n (%) unless stated otherwise

[b] Comorbidity data missing for DWS+ (1) and SD (4).

cases tested positive for DENV by rRT-PCR, and this did not differ between severity categories (Tables 1 and S1). The proportion of DWS-/DWS+ cases with detectable NS1 (77/122, 63.1%) was significantly lower than SD cases (21/23, 91.3%; p = 0.010). DENV-4 was the predominant type, present in 78.5% of the typed samples overall (102/130). Infections by Zika and chikungunya viruses were not detected, nor were coinfections by two DENV serotypes.

Acute-phase samples were tested with two serologic tests: the pGOLD assay for anti-DENV and anti-ZIKV IgM and IgG, and a commercial ELISA for anti-DENV IgM and IgG (Tables 3 and S2). The proportion of individuals with detectable anti-DENV IgM was significantly higher with the pGOLD assay (p<0.001, S3 Table). Although a smaller proportion of SD cases had detectable anti-DENV IgM compared to DWS-/DWS+ cases by either method, this

**Table 2. Binomial logistic regression of participant demographics and disease severity.**

| | SD vs. DWS-/DWS+ | |
|---|---|---|
| Characteristic | OR | 95% CI |
| Age, years | 1.07 | 1.03, 1.11 |
| Gender, female | 0.20 | 0.05, 0.77 |
| Comorbidity, count | 2.09 | 1.26, 3.48 |

Abbreviations: CI, confidence interval; OR, odds ratio

**Table 3. Serologic test results stratified by disease severity.**

| Serologic Test | DWS-/DWS+[a] | SD[a] | p-value |
|---|---|---|---|
| *pGOLD*[b] | | | |
| DENV IgM | 71/119 (59.7) | 7/20 (35.0) | 0.040 |
| DENV IgG | 102/119 (85.7) | 18/20 (90.0) | 1.00 |
| ZIKV IgM | 5/119 (4.2) | 0/20 (0) | 1.00 |
| ZIKV IgG | 19/119 (16.0) | 4/20 (20.0) | 0.52 |
| *ELISA* | | | |
| DENV IgM | 40/122 (32.8) | 5/23 (21.7) | 0.40 |
| DENV IgG | 106/122 (86.9) | 22/23 (95.7) | 1.00 |

[a] Presented as positive/tested (%)

[b] pGOLD testing was performed on 139 participants with sufficient serum available

difference only reached significance for the pGOLD assay. Most participants had detectable anti-DENV IgG by either method: 120/139 (86.3%) in the pGOLD, 128/145 (88.3%) by commercial ELISA. The proportion of individuals with detectable anti-DENV IgG did not differ significantly by severity category (Tables 3 and S2) or test method (p = 0.07, S3 Table).

The pGOLD assay yields a quantitative result that correlates with DENV neutralizing titers (S1 Fig) [59]. In crude binary logistic regression, for every unit increase in anti-DENV IgG, the odds of SD increased by a factor of 2.54 (95% CI, 1.19–5.42). No interaction was observed between anti-DENV IgG and day post-symptom onset, which was not included in the final logistic regression. No association was found between quantitative anti-DENV IgM results and disease severity in crude binary logistic regression.

## Clinical laboratory data

Mean values for most routine laboratory tests, LBP, and chymase differed significantly between DWS-/DWS+ and SD cases (Fig 1, S4 Table). Laboratory values were similar between DWS- and DWS+ cases except for platelet count, which demonstrated a stepwise decrease from DWS- to DWS+ to SD, and serum glutamic oxaloacetic transaminase (SGOT) and LBP, which increased across severity categories (S2 Fig, S5 Table). DENV viral load did not differ by severity category.

Routine laboratory tests were obtained at the discretion of the clinical care team, and as a result, many participants were missing data, particularly for analytes in the metabolic panel (S4 and S5 Tables). Due to this fact, crude associations with SD were calculated for all variables by binomial regression (Table 4), and variables evaluated in the laboratory domain multivariable logistic regression were limited to lymphocyte percent, platelet count, hematocrit, LBP, and chymase. These analytes displayed crude associations with SD, had sufficient data points to maintain model strength, and were not collinear with each other (e.g., hemoglobin and hematocrit, neutrophil and lymphocyte percent). After evaluating these five variables, those with non-significant regression coefficients were removed (LBP and hematocrit). In the final binary logistic regression model, lymphocyte percent, platelet count, and chymase were found to be associated with SD with a very good model fit (AUROC statistic, 0.95; Table 5).

## Chymase and SD

Mean chymase level was significantly higher among individuals with comorbidities (10.75, st. dev. 22.01) compared to those without (2.41, 9.99; p = 0.014). Notably, the single DWS- case with an elevated chymase level (Figs 1K and S1K) occurred in an individual with systemic

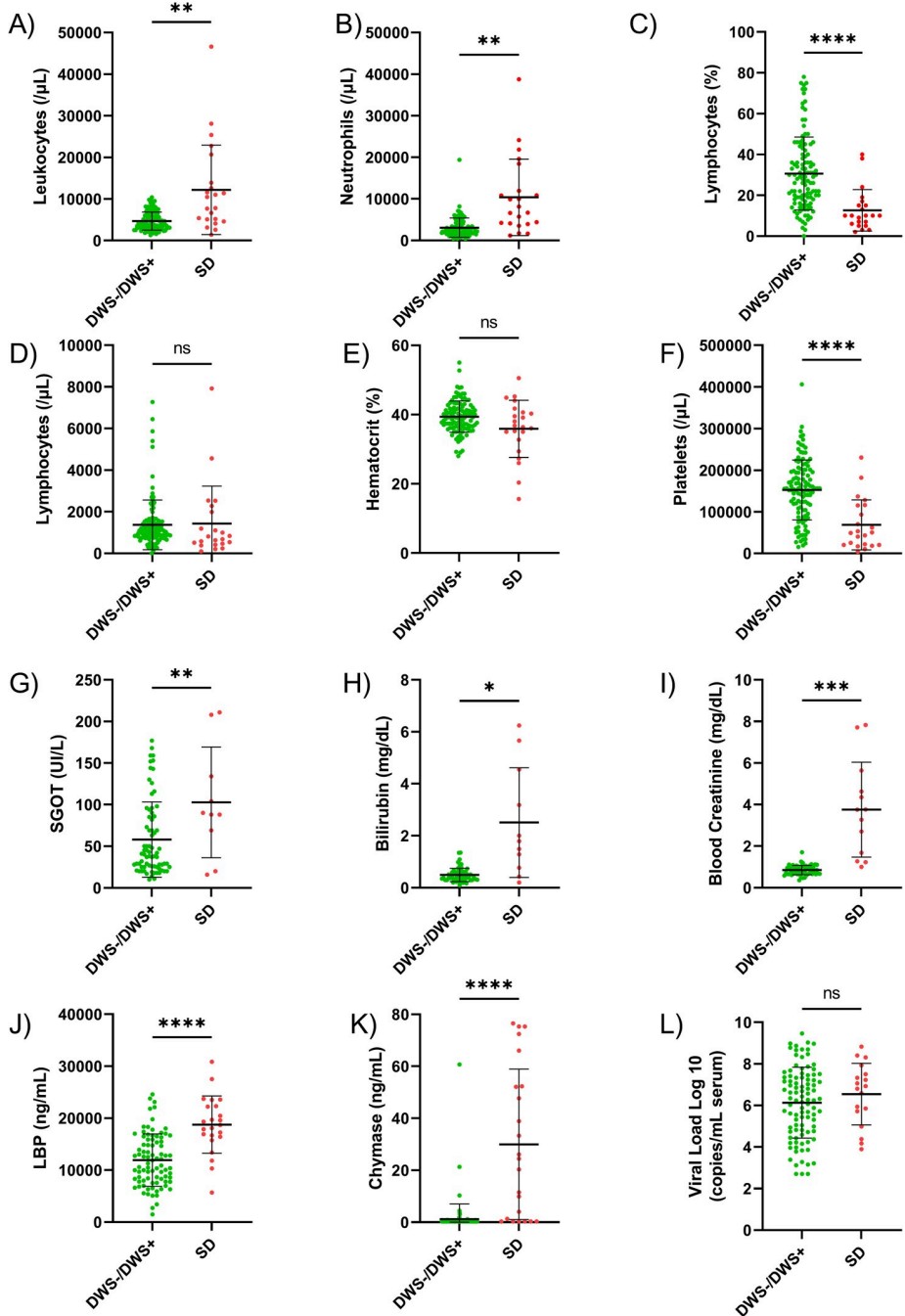

**Fig 1.** **A-I)** Clinical laboratory test result distributions by disease severity. **J-L)** Potential markers of disease severity measured in the current study: J) lipopolysaccharide binding protein (LBP), K) chymase, and L) DENV viral load by disease category. Bars on all graphs represent mean and standard deviation. Labels on the graphs indicate the following: ns, not significant, p>0.05; *, p≤0.05; **, p≤0.01; ***, p≤0.001; ****, p≤0.0001.

lupus erythematosus. To evaluate for a potential interaction between chymase and comorbidities on the development of SD, logistic regression was performed including these two variables with comorbidities defined as a discrete variable counting the number of comorbidities each patient has. Interaction product terms were nonsignificant in binomial and multinomial

**Table 4. Crude associations between laboratory results and dengue severity from binomial logistic regression.**

| Laboratory Value | DWS-/DWS+[a] | SD[a] | OR for SD[b] |
|---|---|---|---|
| Bilirubin (mg/dL) | 0.49±0.26 | 2.51±2.11 | 29.03 (3.74, 225.04) |
| Blood Creatinine (mg/dL) | 0.85±0.22 | 3.76±2.29 | 372.71 (4.38, >999) |
| Chymase (mg/mL) | 1.2±5.8 | 30.0±28.9 | 1.12 (1.06, 1.18) |
| Hematocrit (%) | 39.4±4.5 | 35.9±8.2 | 0.89 (0.82, 0.97) |
| LBP (1,000 ng/mL) | 11,917±5,030 | 18,766±5,510 | 1.28 (1.11, 1.40) |
| Leukocytes (1,000 /μL) | 4,692±2,155 | 12,192±10,765 | 1.43 (1.19, 1.71) |
| Lymphocyte Percentage (%) | 30.6±17.9 | 12.6±10.1 | 0.89 (0.84, 0.94) |
| Platelets (1,000 /μL) | 152,462±72,110 | 68,364±60,249 | 0.98 (0.97, 0.99) |
| SGOT (IU/L) | 58.0±45.2 | 102.8±66.6 | 1.02 (1.00, 1.03) |

[a] Displayed as mean ± standard deviation
[b] Presented as odds ratio for SD (95% CI) vs. DWS-/DWS+

models. Together, comorbidities had an OR of 3.17 (1.71, 5.89) for binomial logistic regression (controlling for chymase); chymase had an OR of 1.11 (1.05, 1.17) (controlling for comorbidities). This model had a strong goodness of fit (C statistic = 0.95).

Anti-NS1 antibodies may modulate chymase release by mast cells in acute dengue. As anti-DENV antibodies detected in the pGOLD assay target whole viral antigen, interactions between chymase and antibodies directed against the NS1 protein of ZIKV were investigated for their association with SD. These antibodies are detected in the pGOLD assay and predominantly represent cross-reactive anti-DENV antibodies in this population [59]. There was no association between chymase level and the quantitative anti-ZIKV NS1 IgM or IgG by linear regression, and no interaction was observed between anti-ZIKV NS1 IgG and chymase in binomial linear regression for SD. However, anti-ZIKV NS1 IgM showed effect modification of chymase in binomial linear regression such that as IgM increased, the chymase OR increased as well. With no detectable anti-ZIKV NS1 IgM, the chymase OR was 1.10 (1.04, 1.17), whereas at the mean level of anti-ZIKV NS1 IgM (0.02 in this population), the chymase OR was 1.21 (1.09, 1.34; AUROC = 0.93).

## Discussion

In a predominantly adult population of dengue cases in Paraguay, due to DENV-1, -2, and -4, multiple factors were associated with clinically severe dengue, including patient (age, gender, comorbidities), serologic (elevated anti-DENV IgG), and laboratory variables (low platelet count, relative lymphopenia, and elevated chymase).

**Table 5. Binomial logistic regression model of laboratory results and disease severity.**

| | SD vs. DWS-/DWS+ | |
|---|---|---|
| Variable | OR | 95% CI |
| Lymphocyte, % | 0.92 | 0.86, 0.98 |
| Platelet count, 1,000/μL[a] | 0.987 | 0.975, 0.999 |
| Chymase, mg/mL | 1.17 | 1.03, 1.33 |

Abbreviations: CI, confidence interval; OR, odds ratio
[a] Three decimal places shown for clarity

Factors identified in the current study are generally consistent with the published dengue literature [4]. Although clinically severe dengue often occurs among children [1, 3, 27], age among adults has been identified as a risk factor for poor outcomes, and in 2019, individuals 15–49 years of age accounted for more deaths and disability adjust life years lost than children [5, 20, 26, 49]. Adults are more likely to develop severe bleeding, and this may be more difficult to manage than plasma leakage that develops in children, for which judicious fluid replacement is often effective [3, 9, 11, 20, 26, 27, 49]. Comorbid illness, including poorly-controlled diabetes mellitus (hemoglobin A1c >7%) and renal disease, have been associated with SD [19, 21], and hypertension has also been identified in certain studies [21]. Notably, in our population, multiple comorbidities were associated with SD, and these demonstrated a cumulative effect when evaluated as a discrete variable. A gender difference among clinically severe dengue has varied across studies [1, 9, 16, 20, 21]. In our population, 66.4% of DWS-/DWS+ cases were female in comparison to only 26.1% of SD cases, and this difference remained significant after controlling for age and comorbidities. Although dengue is often associated with leukopenia [3, 27, 65–67], SD cases in the current study had a mild leukocytosis with reduced lymphocyte percentage (and a resulting neutrophil predominance). Thrombocytopenia is a common finding in SD cases and was one of the few factors that demonstrated a stepwise change across disease severity categories (DWS-, DWS+, and SD) [1, 3, 20, 21, 27, 41, 65].

Chymase and LBP were evaluated as two markers of clinically severe dengue based on data from their use in South and Southeast Asia [4, 34, 35, 37, 38, 41, 45–47]. Both demonstrated a crude association with SD compared to DWS-/DWS+. Although LBP did not remain in the final laboratory domain model, it demonstrated a stepwise increase across the categories of severity, which may have limited power in this study to identify a significant difference in a binomial model. Chymase, along with tryptase and other mast cell degranulation factors, has been associated with clinically severe dengue in several publications [35–38, 68], and the current study confirmed this finding among dengue cases in Paraguay. As clinically severe dengue appears to be more common in Southeast Asia relative to the Americas [2, 69], it is important to study potential differences in pathophysiology between these regions and confirm markers of severity between populations. As markers of dengue severity, mast cell degranulation factors have demonstrated more consistent results that other potential markers such as chemokines and cytokines [34–36, 40, 70], tryptase may play a direct pathophysiologic role in endothelial permeability [68], and mast cells can be stabilized by available FDA-approved medications [35]. Chymase release from mast cells may be modulated by specific anti-DENV antibodies. In mice, pre-treatment with anti-DENV IgG increased chymase release in an FCγRIII-dependent manner [36], and anti-NS1 IgG blocked mast cell degranulation [45]. In the current study, we observed an interaction between chymase level and antibodies against the NS1 protein of ZIKV, a closely related flavivirus. Further evaluation of this interaction using an array of DENV NS1 proteins may delineate a mechanism of protection for anti-NS1 antibodies, which demonstrate epitope-specific protection or enhancement [71, 72].

Higher levels of anti-DENV IgG in the pGOLD multiplex serologic assay were also associated with SD in our study population. This is consistent with findings in secondary dengue cases, though this is difficult to determine with certainty in acute-phase samples [3, 73], and SD can occur in primary dengue, particularly among adults experiencing a first infection [74]. Quantitative anti-DENV IgG levels in the pGOLD assay correlate with DENV $FRNT_{50}$, and we previously observed that higher levels are associated with hospitalization in dengue cases [59]. This finding was confirmed in the current study when applying more consistent criteria for clinically severe dengue [3]. However, simultaneous detection of anti-ZIKV NS1 IgG did not increase the risk for SD in contrast to our earlier findings [59]. Anti-DENV IgM detection in the pGOLD proved more sensitive than a commercial ELISA and demonstrated little cross-

reactivity on the ZIKV NS1 antigen. Notably, interpretation of these results required the use of a control sample that previously tested positive for anti-DENV IgM, and inclusion of a calibrator with this assay would improve generalizability.

DENV serum viral load was not associated with SD in this cross-sectional study. Viral load decreases rapidly over the first week post-symptom onset, and viral kinetics differ between primary and secondary dengue [15, 75–83]. It is therefore difficult to capture peak viremia in most clinical settings. With only a single data point for each patient in our study, the lack of association between viral load and SD is not unexpected, but this highlights a potential limitation of using viral load as a predictor of severity in clinical practice.

Difficulties in studying predictors of clinically severe dengue stem from the low proportion of severe cases among all DENV infections, lack of rapid and accurate diagnostics, and variability in the definition of study endpoints [3, 8, 9, 13]. The current study relied principally on DENV rRT-PCR for diagnosis, with a subset of participants detected by NS1. As part of the parent study design, participants typically presented with fever, which may bias this group toward more severe cases [52, 84]. Nonetheless, seven factors were associated with clinically severe dengue: five of these are commonly available at the acute visit (age, gender, comorbidities, platelet count, and lymphocyte percentage) and chymase and anti-DENV IgG can be measured by ELISA. Study designs that enroll participants based on rapid antigen test results limit the sample size necessary to include enough severe cases, but this may bias the study population given the clinical performance of current rapid tests [3, 28, 52, 85, 86]. An improved antigen diagnostic in combination with a prognostic test may then increase DENV detection, identify individuals at high risk for SD, and facilitate future trials for clinically severe dengue.

This study had several limitations. First, a single acute-phase sample was available for each participant. Samples were obtained at different timepoints in relation to the development of severe disease among the participants, such that the study was not designed to prospectively evaluate each marker as a predictor of clinically severe dengue. Second, although all available SD cases were included, the sample size was small, particularly for the detection of differences among factors with relatively narrow value ranges, such as quantitative pGOLD values. Third, routine labs were collected at the discretion of the care team, and as a result, not all participants had laboratory values within the correct time frame. This limited the variables included in the laboratory domain multivariable analysis.

Although dengue is endemic in Paraguay, scarce studies evaluate severity markers in this population. Therefore, our findings may aid in the early detection of potentially severe dengue cases and serve as a basis for the development of new combined diagnostic-prognostics for use in acute-phase and serial samples from dengue cases.

## Supporting information

**S1 Fig. Distributions of DENV pGOLD serology results by disease severity.**
(PDF)

**S2 Fig. Distribution of selected laboratory test results by disease severity.**
(PDF)

**S1 Table. Characteristics of participants by severity category.**
(PDF)

**S2 Table. Serologic test results stratified by disease severity.**
(PDF)

**S3 Table. Comparison of pGOLD and Vircell ELISA detection of anti-DENV IgM and IgG in acute-phase samples.**
(PDF)

**S4 Table. Laboratory values by disease severity.**
(PDF)

**S5 Table. Laboratory values by disease severity categories.**
(PDF)

## Acknowledgments

We thank the members of the study team based at the Instituto de Investigaciones en Ciencias de la Salud, Universidad Nacional de Asunción; Hospital Villa Elisa; and Hospital Central del Instituto de Provisón Social, all located in Paraguay, for their dedication to this study and their excellent work. We are grateful to the study participants and their families.

## Author Contributions

**Conceptualization:** Cynthia Bernal, Alejandra Rojas, Yvalena de Guillén, Patricia Langjahr, Benjamin A. Pinsky, Fátima Cardozo, Jesse J. Waggoner.

**Data curation:** Cynthia Bernal, Alejandra Rojas, Oliver Caballero, Victoria Stittleburg, Fátima Cardozo, Jesse J. Waggoner.

**Formal analysis:** Cynthia Bernal, Sara Ping, Alejandra Rojas, Victoria Stittleburg, Benjamin A. Pinsky, Fátima Cardozo, Jesse J. Waggoner.

**Funding acquisition:** Cynthia Bernal, Alejandra Rojas, Benjamin A. Pinsky, Fátima Cardozo, Jesse J. Waggoner.

**Investigation:** Cynthia Bernal, Alejandra Rojas, Oliver Caballero, Victoria Stittleburg, Marta Von-Horoch, Patricia Luraschi, Sandra Cabral, María Cecilia Sánchez, Aurelia Torres, Fátima Cardozo, Jesse J. Waggoner.

**Methodology:** Cynthia Bernal, Sara Ping, Alejandra Rojas, Oliver Caballero, Victoria Stittleburg, Marta Von-Horoch, Patricia Luraschi, Sandra Cabral, María Cecilia Sánchez, Aurelia Torres, Fátima Cardozo, Jesse J. Waggoner.

**Project administration:** Alejandra Rojas, Yvalena de Guillén, Patricia Langjahr, Marta Von-Horoch, Patricia Luraschi, Sandra Cabral, María Cecilia Sánchez, Aurelia Torres, Fátima Cardozo, Jesse J. Waggoner.

**Resources:** Sara Ping, Alejandra Rojas, Yvalena de Guillén, Patricia Langjahr, Fátima Cardozo, Jesse J. Waggoner.

**Software:** Cynthia Bernal, Sara Ping, Alejandra Rojas, Oliver Caballero, Victoria Stittleburg, Fátima Cardozo, Jesse J. Waggoner.

**Supervision:** Alejandra Rojas, Yvalena de Guillén, Patricia Langjahr, Benjamin A. Pinsky, Marta Von-Horoch, Patricia Luraschi, Sandra Cabral, María Cecilia Sánchez, Aurelia Torres, Fátima Cardozo, Jesse J. Waggoner.

**Validation:** Cynthia Bernal, Sara Ping, Alejandra Rojas, Fátima Cardozo, Jesse J. Waggoner.

**Visualization:** Cynthia Bernal, Sara Ping, Alejandra Rojas, Benjamin A. Pinsky, Fátima Cardozo, Jesse J. Waggoner.

**Writing – original draft:** Cynthia Bernal, Sara Ping, Alejandra Rojas, Oliver Caballero, Yvalena de Guillén, Patricia Langjahr, Benjamin A. Pinsky, Marta Von-Horoch, Patricia Luraschi, Sandra Cabral, María Cecilia Sánchez, Aurelia Torres, Fátima Cardozo, Jesse J. Waggoner.

**Writing – review & editing:** Cynthia Bernal, Sara Ping, Alejandra Rojas, Oliver Caballero, Victoria Stittleburg, Yvalena de Guillén, Patricia Langjahr, Benjamin A. Pinsky, Marta Von-Horoch, Patricia Luraschi, Sandra Cabral, María Cecilia Sánchez, Aurelia Torres, Fátima Cardozo, Jesse J. Waggoner.

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
