## [Decision Letter · Decision Letter 0]

2 Oct 2022

Dear Dr. Waggoner,

Thank you very much for submitting your manuscript "Serum biomarkers and anti-flavivirus antibodies at presentation as indicators of severe dengue" for consideration at PLOS Neglected Tropical Diseases. As with all papers reviewed by the journal, your manuscript was reviewed by members of the editorial board and by several independent reviewers. In light of the reviews (below this email), we would like to invite the resubmission of a significantly-revised version that takes into account the reviewers' comments. 

We cannot make any decision about publication until we have seen the revised manuscript and your response to the reviewers' comments. Your revised manuscript is also likely to be sent to reviewers for further evaluation.

Sincerely,

Michael Benjamin Arndt 

Academic Editor

Abdallah Samy

Section Editor

Reviewer's Responses to Questions

**Key Review Criteria Required for Acceptance?**

**Methods**

-Are the objectives of the study clearly articulated with a clear testable hypothesis stated?

-Is the study design appropriate to address the stated objectives?

-Is the population clearly described and appropriate for the hypothesis being tested?

-Is the sample size sufficient to ensure adequate power to address the hypothesis being tested?

-Were correct statistical analysis used to support conclusions?

-Are there concerns about ethical or regulatory requirements being met?

Reviewer #1: Rather than using the brand name of the methods in the abstract “pGold assay”, please write the type of assay used so it is clear what information was gained without reading the methods section. (e.g. DENV and ZIKV IgG/IgM?) The brand name is more appropriate for the methods section. 

The aspect of defining dengue based on “strong epidemiological link” is unclear and maybe a weak definition depending on the reasoning. At least this is non-standard. More information is needed here to fully consider this inclusion criteria. 

It is also unclear in the methods if the collection of “acute” samples preceded the final diagnosis involving severity? Even if not, it should be stated clearly. 

Is it not odd that most of the cases are DENV IgM-negative when this should be induced during acute dengue, including acute secondary dengue? 

The discussion does state that this was "not a prospective study", but it would be important to clearly define it as a retrospective study in the methods, if appropriate.

Reviewer #2: - The objective is to evaluate the biomarkers of dengue severity at a single timepoint. The study is exploratory and no specific hypotheses were stated. 

- This cross-sectional observational cohort is appropriate for an exploratory study. 

- The population was generally clearly described, but I do have a few follow-up questions on the population. 

1) In line 154, authors mention that samples were tested for ZIKV, chikungunya and DENV, but they do not state if any subjects tested positive for ZIKV and/or CHIKV. Can they clarify whether any participants had coinfections and if so, how coinfections were managed?

2) In line 142, they state that all SD cases were included, and a mixture of DWS- and DWS+ cases was then selected to achieve a representative distribution of participants. Can they state how many individuals were in the full cohort? And, how did they select participants for the subcohort? Was a statistical package used for this?

- Since this was exploratory, they did not include power calculations. However, can they state why they chose n = 55 participants with DWS- and n = 67 participants with DWS+? Was there some significance to these numbers? 

- statistical analyses seemed appropriate and were relatively easy to follow

- no ethical concerns

One other question I had about the methods was can they explain how they created the comorbidity variable included in the models? It looks like in the model described in table 2, the comorbidity variable was dichotomous: present/ absent. Was comorbidities also coded as a dichotomous variable in the chymase interaction model (lines 287 – 296)?

Reviewer #3: Methods: How were these 145 participants selected from the “parent” study? How many subjects are in the parent study, and what criteria did you use to select 145 out of the total number? Was it consecutive recruitment, or selection based on certain inclusion criteria that were different to the inclusion criteria in the parent study?

Why was chymase selected and not also tryptase?

**Results**

-Does the analysis presented match the analysis plan?

-Are the results clearly and completely presented?

-Are the figures (Tables, Images) of sufficient quality for clarity?

Reviewer #1: Line 111- are these anti-DENV and anti-ZIKV binding titers or are they neutralizing titers? Because DENV antibody cross-reactivity can lead to positive signals for ZIKV even in populations that have not been exposed to the virus. Some clarification would be helpful here. 

The assessment of anti-NS1 IgM levels is very interesting but did the authors measure NS1 levels itself? 

For those patients who are ZIKV NS1-IgM positive, do they have neutralizing antibodies against ZIKV? i.e. do they just have more cross-reactive DENV antibodies or are they truly ZIKV-immune? 

Could the authors please add the data controlling for day post-sympom onset (data not shown)? It seems clear that time should be a co-variate. 

Figure 1 Panel e- The scale of the y-axis appears wrong. Please check all units/scales

Reviewer #2: - The exploratory analysis presented matches the plan

- Results are well-written, tables and figures are clear and easy to follow

I do have a few comments and questions about the results:

1) From the models, it looks like the effect size of comorbidities is quite large, and I think it would be valuable if the authors could delve into this variable a bit more. Specifically, from the table 1 footnote, it looks like 32 had hypertension, 14 had diabetes and 30 had other. What were the other comorbidities? It looks like the models may have just considered presence/absent of comorbidities, but were there differences in the number of comorbidities between people with SD vs DWS-/DWS+? Was a specific comorbidity more predictive? For example, did all participants with SD have diabetes but an equal number of participants in both groups have hypertension? From a clinical perspective, it is nice to know what comorbidities to look out for, and if your patient does have a certain comorbidity, how might that relate to the presence of SD? 

2) Line 248: can authors clarify does the pGOLD assay correlates with DENV1-4 neutralizing titers (i.e. geometric mean titer)? Or does it correlate with the neutralizing titers against a DENV2 since it is based on antibodies against DENV2?

3) Line 274: In their model presented in Table 5, I wasn’t sure why the authors focused on lymphocyte percentage when there was no actual difference in lymphocyte count. In contrast, there was a difference in both neutrophil count and percentage between the two groups. So, in this cohort, aren’t the differences in neutrophils really driving the observations, and the difference in lymphocyte percentage is just a response to difference in neutrophils? For example, if there are more neutrophils, then the neutrophil percentage goes up, which pushes the lymphocyte percentage down. Can the authors clarify why they focused on lymphocyte percentage rather than neutrophil count and percentage?

- Additionally, can the authors explain why they picked lymphocyte percent, platelet count, and chymase for table 5? I understand that bilirubin, SGOT and creatinine had many missing values but what about LBP, neutrophils, hematocrit? I didn’t quite understand why the authors went from a multivariate logistic regression with multiple variables (line 271) to a logistic regression with 3 variables (line 274-275). Did they see something in the multivariate regression that motivated the smaller model? 

4) Line 298 – 307: can the authors explain why they chose to look for interactions between ZIKA IgM and IgG and chymase? It seemed that Zika IgM was associated with chymase OR, but none of the patients with severe dengue had a positive Zika IgM. In contrast, they state that anti-DENV IgG levels were higher in those with severe dengue (line 349). Additionally, they state that Dengue IgG is associated with increased chymase in other studies (line 88-89 and line 342). Given these findings, wouldn’t it make sense to also look for associations between anti-DENV IgG and chymase? 

5) In Line 349, they state that higher levels of anti-DENV IgG and undetectable anti-DENV IgM were associated with SD in their population. However, Table 3 doesn’t really show these relative differences, especially in IgG levels. Instead, it only shows whether the antibodies were present or not. Instead of a table, it might be more informative to have a figure with serology distributions for each group, as is shown for the laboratory values in figure 1. 

6) Table S4 and S5 have a superscript a that says “missing data,” but that superscript does not seem to have any information.

Reviewer #3: Nice tables!

Tables: shouldn`t abbreviations such as LBP not be spelled out in a legend?

**Conclusions**

-Are the conclusions supported by the data presented?

-Are the limitations of analysis clearly described?

-Do the authors discuss how these data can be helpful to advance our understanding of the topic under study?

-Is public health relevance addressed?

Reviewer #1: The abstract should clarify whether the selected cases for biomarkers were confirmed dengue. As written currently, it suggests that all “suspected arboviral infections” were used to model dengue. 

Line 86: “Chymase release from mast cells occurs in the presence of DENV and may be increased by pre-existing anti-DENV IgG antibodies or blocked by antibodies against viral non-structural protein 1 (NS1) [33, 42]. The first part of this sentence has been shown in humans and mice, whereas the second part relies on a single citation from an immune compromised mouse study. This should be clarified since the second aspect in mice has not been reproduced independently or shown in humans or with human cells. 

Line 289, this is consistent with citation 34 as well.

Line 343- Is model system indicated correct? Is it not in mice?

This is a very controversial statement (Line 380) "However, as many pathways associated with clinically severe dengue appear linked to NS1, including chymase release” Actually only one study suggests an association between NS1 and chymase release, whereas most studies examined the role of the virus itself or antibody-mediated triggering of mast cells by immune complexes. In fact, in many studies NS1 levels do not correlate with disease severity even when chymase levels do. It is plausible that NS1-antibodies could contribute to mast cell activation during dengue but it’s not well supported and even incorrect to suggest this is the only mechanism.

Reviewer #2: I thought the conclusions were clear and fairly written. I appreciated that they highlighted potential reasons for not observing differences in viral load between groups, and the discussions of the limitations were thorough and helpful. 

-Line 398: can they really say that these findings will aid in the early detection of potentially severe dengue cases? They did not state that individuals with severe dengue were particularly early in their course when they had these abnormalities. I think the most they can conclude is that these variables might aid in early detection and should be evaluated in prospective trials.

Reviewer #3: What is the significance of your findings? How should they translate into clinical practice? We already know that chymase, platelets and lymphocytes are predictive, so what is the added value of your study?

**Editorial and Data Presentation Modifications?**

Reviewer #1: There are many occasions where the writing is a bit confusing and there could be errors in the text that need to be checked carefully, but overall the data appears to be well presented and clear.

Reviewer #2: (No Response)

Reviewer #3: Abstract : Most readers would not know where Asuncion is, please add country.

Introduction: please refer to this recent review that provides an updated mapping of dengue globally over the past 30 years: 

Global burden for dengue and the evolving pattern in the past 30 years. 

Yang X, Quam MBM, Zhang T, Sang S. J Travel Med. 2021 Dec 29;28(8):taab146. doi: 10.1093/jtm/taab146. PMID: 34510205 

Also important to highlight that dengue is not only a problem of the endemic population but also increasingly of the travellers population:

Dengue, chikungunya and Zika in GeoSentinel surveillance of international travellers: a literature review from 1995 to 2020. 

Osman S, Preet R. J Travel Med. 2020 Dec 23;27(8):taaa222. doi: 10.1093/jtm/taaa222. PMID: 33258476 

Maybe also highlight that contrary to common perception, also primary dengue can have fatal outcomes: 

Fatal outcomes of imported dengue fever in adult travelers from non-endemic areas are associated with primary infections. 

Huits R, Schwartz E. J Travel Med. 2021 Jul 7;28(5):taab020. doi: 10.1093/jtm/taab020.

Discussion: The authors should ideally also set their chymase results in relation to tryptase, and other biomarkers. A systematic review on biomarkers was published. 

Markers of dengue severity: a systematic review of cytokines and chemokines. 

Lee YH, Leong WY, Wilder-Smith A. J Gen Virol. 2016 Dec;97(12):3103-3119. doi: 10.1099/jgv.0.000637. Epub 2016 Oct 21. PMID: 27902364 

Dengue virus-elicited tryptase induces endothelial permeability and shock. 

Rathore AP, Mantri CK, Aman SA, Syenina A, Ooi J, Jagaraj CJ, Goh CC, Tissera H, Wilder-Smith A, Ng LG, Gubler DJ, St John AL. J Clin Invest. 2019 Jul 2;129(10):4180-4193. doi: 10.1172/JCI128426. PMID: 31265436 

References: Ref 1 is now quite old and could be updated by a similar more recent paper in the same journal with the same title:

Dengue. 

Lancet. 2019 Jan 26;393(10169):350-363. doi: 10.1016/S0140-6736(18)32560-1.

**Summary and General Comments**

Reviewer #1: This is a valuable study that reproduces some of the key biomarkers of severity that have been identified in prior literature, independently, in a new dengue cohort. It is equally important to share the results where the authors' data is negative, showing a statistical difference between groups but poor predictive value in their model. However, in the absence of mechanistic data the authors have relied on interpretations from the literature for discussion, which sometimes are not well supported. This study could be brought to publishable form by improving the writing and description of the methods, adding some data that was not shown (e.g. NS1 concentrations, time based analyses), and toning down over-interpretations relating to mechanisms of disease that are assumed based on other literature but not explicitly proven here.

Reviewer #2: Overall, this is an interesting and clinically relevant exploratory study aiming to identify biomarkers of dengue severity at a single time-point. It is valuable to read about severe dengue and DENV4 cases in Latin America, both of which are relatively rare in the literature. The manuscript is well-written and easy to follow. My questions are listed in each individual section, but to summarize, they include:

1. What is the size of their full cohort and how did they choose their subcohort of participants with DWS-/DWS+?

2. What comorbidities did they observe, and were the number of comorbidities and/or certain comorbidities more important than others? 

3. Could they present table 3 as a figure with distributions of each serology, so readers can see the relative differences, such as those in DENV-IgG levels?

4. Why did they focus on lymphocyte percentage as a predictive variable when differences in neutrophils counts seemed to be driving the observations?

5. Why did they test for a relationship between anti-ZIKV IgM and IgG and chymase but not anti-DENV IgM and IgG and chymase?

Reviewer #3: This study aims to identify markers for the early detection of potentially severe dengue cases and inform the development of new prognostics for use in acute-phase and serial samples. The results are not entirely new as decreasing platelet and increasing chymase and low lymphocytes have previously been reported to be predictive of severe dengue. 

Limitations are the small sample size. The strength is the inclusion of a wide age range in order to identify age as a risk factor. It is also unique in that the cohort is a predominantly adult population whilst most studies are generally done in pediatric cohorts. 

The conclusions should therefore be stronger: maybe a point of care test should be developed to test for chymase to help triage patients in the first days of illness?

Overall, the manuscript is very well written, highlights its limitations and the figures and tables are well done. 

Abstract : Most readers would not know where Asuncion is, please add country.

Introduction: please refer to this recent review that provides an updated mapping of dengue globally over the past 30 years: 

Global burden for dengue and the evolving pattern in the past 30 years. 

Yang X, Quam MBM, Zhang T, Sang S. J Travel Med. 2021 Dec 29;28(8):taab146. doi: 10.1093/jtm/taab146. PMID: 34510205 

Also important to highlight that dengue is not only a problem of the endemic population but also increasingly of the travellers population:

Dengue, chikungunya and Zika in GeoSentinel surveillance of international travellers: a literature review from 1995 to 2020. 

Osman S, Preet R. J Travel Med. 2020 Dec 23;27(8):taaa222. doi: 10.1093/jtm/taaa222. PMID: 33258476 

Maybe also highlight that contrary to common perception, also primary dengue can have fatal outcomes: 

Fatal outcomes of imported dengue fever in adult travelers from non-endemic areas are associated with primary infections. 

Huits R, Schwartz E. J Travel Med. 2021 Jul 7;28(5):taab020. doi: 10.1093/jtm/taab020.

Methods: How were these 145 participants selected from the “parent” study? How many subjects are in the parent study, and what criteria did you use to select 145 out of the total number? Was it consecutive recruitment, or selection based on certain inclusion criteria that were different to the inclusion criteria in the parent study?

Why was chymase selected and not also tryptase?

Tables: shouldn`t abbreviations such as LBP not be spelled out in a legend?

Discussion: The authors should ideally also set their chymase results in relation to tryptase, and other biomarkers. A systematic review on biomarkers was published. 

Markers of dengue severity: a systematic review of cytokines and chemokines. 

Lee YH, Leong WY, Wilder-Smith A. J Gen Virol. 2016 Dec;97(12):3103-3119. doi: 10.1099/jgv.0.000637. Epub 2016 Oct 21. PMID: 27902364 

Dengue virus-elicited tryptase induces endothelial permeability and shock. 

Rathore AP, Mantri CK, Aman SA, Syenina A, Ooi J, Jagaraj CJ, Goh CC, Tissera H, Wilder-Smith A, Ng LG, Gubler DJ, St John AL. J Clin Invest. 2019 Jul 2;129(10):4180-4193. doi: 10.1172/JCI128426. PMID: 31265436 

References: Ref 1 is now quite old and could be updated by a similar more recent paper in the same journal with the same title:

Dengue. 

Lancet. 2019 Jan 26;393(10169):350-363. doi: 10.1016/S0140-6736(18)32560-1.

PLOS authors have the option to publish the peer review history of their article (what does this mean?). If published, this will include your full peer review and any attached files.

Reviewer #1: Yes: Ashley St. John

Reviewer #2: Yes: Camila D. Odio

Reviewer #3: No
---

## [Decision Letter · Decision Letter 1]

10 Feb 2023

Dear Dr. Waggoner,

We are pleased to inform you that your manuscript 'Serum biomarkers and anti-flavivirus antibodies at presentation as indicators of severe dengue' has been provisionally accepted for publication in PLOS Neglected Tropical Diseases.

Best regards,

Michael Benjamin Arndt, PhD, MPH

Academic Editor

Abdallah Samy

Section Editor

Reviewer's Responses to Questions

**Key Review Criteria Required for Acceptance?**

**Methods**

-Are the objectives of the study clearly articulated with a clear testable hypothesis stated?

-Is the study design appropriate to address the stated objectives?

-Is the population clearly described and appropriate for the hypothesis being tested?

-Is the sample size sufficient to ensure adequate power to address the hypothesis being tested?

-Were correct statistical analysis used to support conclusions?

-Are there concerns about ethical or regulatory requirements being met?

Reviewer #1: The authors addressed all of my questions.

Reviewer #2: (No Response)

**Results**

-Does the analysis presented match the analysis plan?

-Are the results clearly and completely presented?

-Are the figures (Tables, Images) of sufficient quality for clarity?

Reviewer #1: All concerns in the results section were addressed

Reviewer #2: (No Response)

**Conclusions**

-Are the conclusions supported by the data presented?

-Are the limitations of analysis clearly described?

-Do the authors discuss how these data can be helpful to advance our understanding of the topic under study?

-Is public health relevance addressed?

Reviewer #1: Conclusions appear appropriate with caveats discussed.

Reviewer #2: (No Response)

**Editorial and Data Presentation Modifications?**

Reviewer #1: None.

Reviewer #2: (No Response)

**Summary and General Comments**

Reviewer #1: This manuscript has improved with revision and I have no further concerns.

Reviewer #2: The authors did a thorough job editing the manuscript per the reviewer recommendations, and the manuscript is appropriate for publication. I think this will be a valuable addition to the literature, especially highlighting severe DENV4 in adults in the Americas.

PLOS authors have the option to publish the peer review history of their article (what does this mean?). If published, this will include your full peer review and any attached files.

Reviewer #1: **Yes: **Ashley St. John

Reviewer #2: **Yes: **Camila D. Odio

---

## [Editor Report · Acceptance letter]

22 Feb 2023

Dear Dr. Waggoner,

We are delighted to inform you that your manuscript, "Serum biomarkers and anti-flavivirus antibodies at presentation as indicators of severe dengue," has been formally accepted for publication in PLOS Neglected Tropical Diseases.

Best regards,

Shaden Kamhawi

co-Editor-in-Chief

Paul Brindley

co-Editor-in-Chief
